# Implementation of an antibiotic resistance surveillance tool in Madagascar, the TSARA project: a prospective, observational, multicentre, hospital-based study protocol

Christelle Elias  ,[1,2] Mathieu Raad,[3] Saida Rasoanandrasana,[4] Antso Hasina Raherinandrasana,[5] Volatiana Andriananja,[6] Mihaja Raberahona  ,[6] Catrin E Moore,[7] Mamy Randria,[4] Laurent Raskine,[3] Philippe Vanhems,[1,2] François-Xavier Babin[3]

For numbered affiliations see end of article.

**Correspondence to**
Dr Christelle Elias;
christelle.elias@chu-lyon.fr

## ABSTRACT

**Introduction** Antimicrobial resistance (AMR) has become a significant public health threat. Without any interventions, it has been modelled that AMR will account for an estimated 10 million deaths annually by 2050, this mainly affects low/middle-income countries. AMR has a systemic negative perspective affecting the overall healthcare system down to the patient's personal outcome. In response to this issue, the WHO urged countries to provide antimicrobial stewardship programmes (ASPs). ASPs in hospitals are a vital component of national action plans for AMR, and have been shown to significantly reduce AMR, in particular in low-income countries such as Madagascar.

As part of an ASP, AMR surveillance provides essential information needed to guide medical practice. We developed an AMR surveillance tool—Technique de Surveillance Actualisée de la Résistance aux Antimicrobiens (TSARA)—with the support of the Mérieux Foundation. TSARA combines bacteriological and clinical information to provide a better understanding of the scope and the effects of AMR in Madagascar, where no such surveillance tool exists.

**Methods and analysis** A prospective, observational, hospital-based study was carried out for data collection using a standardised data collection tool, called TSARA deployed in 2023 in 10 hospitals in Madagascar participating in the national Malagasy laboratory network (Réseau des Laboratoires à Madagascar (RESAMAD)). Any hospitalised patient where the clinician decided to take a bacterial sample is included. As a prospective study, individual isolate-level data and antimicrobial susceptibility information on pathogens were collected routinely from the bacteriology laboratory and compiled with clinical information retrieved from face-to-face interviews with the patient and completed using medical records where necessary. Analysis of the local ecology, resistance rates and antibiotic prescription patterns were collected.

**Ethics and dissemination** This protocol obtained ethical approval from the Malagasy Ethical Committee n°07-MSANP/SG/AGMED/CNPV/CERBM on 24 January 2023.

---

**STRENGTHS AND LIMITATIONS OF THIS STUDY**

⇒ Technique de Surveillance Actualisée de la Résistance aux Antimicrobiens (TSARA) is a prospective, observational and multicentre protocol capturing data from hospitalised patients.
⇒ TSARA is an antimicrobial resistance surveillance tool that compiles microbiological and clinical data, including information on antibiotic prescriptions implemented in Malagasy hospitals belonging to the Réseau des Laboratoires à Madagascar network.
⇒ The data collection is performed on an electronic tablet which limits data entry bias.
⇒ The TSARA project is highly dependent on the infrastructure and the available resources in Malagasy laboratories.

---

Findings generated were shared with national health stakeholders, microbiologists, members of the RESAMAD network and the Malagasy academic society of infectious diseases.

## INTRODUCTION

Antimicrobial resistance (AMR) is a global health threat associated with a poor prognosis and an increased mortality in the population infected with resistant bacteria. In the coming decades, deaths attributable to AMR will overcome the mortality due to non-communicable diseases through the lack of treatment options.[1] Recent data from the Global Burden of Disease[2] suggested that the impact of AMR on health is even more important than previous estimations.[1] To address the increasing problem, the WHO published the Global Action Plan on AMR in 2015[3] and urged countries to build national action plans aiming to prioritise

interventions to combat AMR rapidly. One of the objectives was to implement and strengthen surveillance systems to monitor AMR as a first step of an antimicrobial stewardship programme (ASP),[4] in particular in hospital settings where high levels of resistance occur. In addition to these measures, the WHO also advocates for the implementation of antibiotic management policies adapted to local resistance patterns in hospitals.[5–8] Setting up an ASP is challenging, especially in low/middle-income countries (LMICs) due to the lack of infrastructure, good quality data and governance.[7 9] For a country like Madagascar, where robust healthcare facilities, infrastructures and laboratories are very scarce, collecting resistance data remains a challenge.

Madagascar is the fifth largest island in the world, but is considered a low-income country according to the World Bank.[10] Health coverage in Madagascar is extremely limited and the patient has to directly pay for the majority of healthcare (ie, diagnostics, treatment, consumables, surgery). Most drugs—especially antibiotics—are dispensed only with a medical prescription. However, in Madagascar, it is very common to deliver such medicines 'Over The Counter' and antibiotics can be sold easily outside hospital settings, for those who can afford them, without any medical prescription. Lack or substandard infection control practices contribute significantly to the spread of bacterial resistance, both in the community (through AMR/antibiotics in wastewater) and in hospitals (hospital hygiene and nosocomial transmission). The culture of systematic treatment against an infectious agent encourages the misuse and the overconsumption of antibiotics, prescribed or not. At the hospital level, AMR has a greater effect, as patients are in hospital for several days, receive more care, have invasive procedures performed and thus are more at risk of transmission of bacteria and healthcare-associated infections. Appropriate hygiene measures

and standard precautions are difficult to implement in low-resource settings, where there is for instance an average of five hospital beds per room in a Malagasy university hospital.

Outside of the Réseau des Laboratoires à Madagascar (RESAMAD) laboratories, there are very few laboratories that can perform microbiology cultures and antimicrobial susceptibility testing (AST) in Madagascar and the burdened cost remains prohibitive for many patients. Targeted antibiotic therapy is therefore rarely prescribed and antibiotic treatment often remains empirical without any de-escalation. In this context of low access to microbiology laboratory combined with high rates of bacterial resistance at the national level, practitioners use broad-spectrum therapies, increasing the selection pressure on the bacteria and creating emerging resistances. Knowledge of the local resistance patterns is essential to adapt the clinical decision-making and have locally adapted recommendations.[8] Table 1 provides a Strengths, Weaknesses, Opportunities and Threats analysis as an overview of what is currently in place and the starting points of the implementation of an AMR surveillance tool in Madagascar.

In Madagascar as well as in multiple developing countries, the Mérieux Foundation has been strengthening clinical laboratories in the fight against infectious diseases for decades to enhance the quality and access to clinical laboratories aiming at preventing infections and improving patient care.[11] The Mérieux Foundation has been investing on the fight against AMR supporting countries to develop infrastructures, laboratories, personnel training and building a laboratory network to strengthen laboratory-based surveillance of infectious diseases. As a result, in 2022, 14 Malagasy laboratories were in the RESAMAD laboratory network, they were all able to perform bacteriology diagnosis, this was created in 2007 and has risen to 27 laboratories across the country to date

**Table 1** SWOT analysis of the implementation of an AMR surveillance tool in hospitals in Madagascar

| Strengths | Weaknesses |
|---|---|
| ► Laboratory network RESAMAD[16] currently in place, already transferring data to the WHO through the GLASS-AMR module<br>► Active support of international stakeholders<br>► Identification of highly motivated Malagasy practitioners willing to work on AMR and ASP<br>► Existence of several scientific societies (infectious diseases, paediatrics, medical biology) | ► Little knowledge on the local epidemiology of AMR<br>► Collection of antibiograms not performed on a daily basis<br>► Lack of visibility on antibiotic prescriptions in hospitals and their efficiency<br>► Implementation only doable where good laboratories exist<br>► Bacteriology results sometimes delivered to the prescriber in a paper format |
| **Opportunities** | **Threats** |
| ► Capture high-level antibiotic use in hospitals<br>► Existing antibiotic recommendations in place, but they are not adapted to local resistance patterns[21]<br>► Existing AMR protocols including the surveillance of antimicrobial use | ► Challenging implementation<br>► Substantial missing data<br>► Personnel require training<br>► Lack of funding<br>► Political instability<br>► Climate disorders |

AMR, antimicrobial resistance; RESAMAD, Réseau des Laboratoires à Madagascar; SWOT, Strengths, Weaknesses, Opportunities and Threats.

(RESAMAD—Fondation Mérieux (fondation-merieux. org)).

Given this context, and the challenge to combat AMR in healthcare settings, ASP in hospitals are a vital component of national action plans for AMR, and have been shown to significantly reduce AMR, in particular in low-income country such as Madagascar.[9 12] As part of an ASP, AMR surveillance provides essential information needed to guide medical practice. An AMR surveillance tool called TSARA (Technique de Surveillance Actualisée de la Résistance aux Antimicrobiens) was developed thanks to the support of the Mérieux Foundation, with the objective of combining patient data with laboratory and epidemiological surveillance data to provide a better understanding of the scope and the effects of AMR in Madagascar, where no such surveillance tool was available, in order to treat patients more effectively. It will allow monitoring and follow-up compared with the scarce existing prescribing data.[13] The TSARA project also aligns with the antimicrobial surveillance programme led by the direction of epidemiological surveillance and response of the Ministry of Health in the context of One Health and in collaboration with other Ministries to tackle AMR in Madagascar. TSARA also answers one of the objectives of the Malagasy national action plan on AMR.[14]

## OBJECTIVES

One of the most important benefits of AMR surveillance at the pathogen level is to achieve a standardised, comparable and easily validated data collection method for quality data that is used to implement better patient treatment and hospital/national guidelines.[15] The data allows an optimisation of the patient care by adapting treatment guidelines according to the patient demographics and type of infection, taking into consideration of the circulating pathogens and local resistance patterns. The primary objective is:

1. To institute a systematic approach towards microbiological data and clinical data collection and analysis of antimicrobial susceptibility information to optimise antimicrobial therapy in Malagasy hospitals.
   The secondary objectives are:
2. To establish timely reporting of information on pathogens of interest and their antimicrobial susceptibility profiles to medical doctors.
3. To allow monitoring of the hospital antibiotic prescription (quantity, quality and spectrum, concordance with bacteriological results).
4. To provide data to the laboratory quality assessment and identify areas for improvement.

## METHODS AND ANALYSIS
### Study design and setting
#### Study design

TSARA is a prospective, observational, multicentre, hospital-based study based on a standardised data collection tool deployed in 10 facilities with both adult and paediatric patients in Madagascar (figure 1), collecting microbiological, demographic, prescription and clinical data in one place.

### Outcome measures

The outcomes of interest will be categorised into behavioural, clinical and microbiological outcomes. Behavioural outcomes include changes in prescribing practices among prescribers and compliance with AMS protocols. In the TSARA project, we consider an adapted antibiotic prescription as a behavioural outcome, defined as:

1. A prescription of an antibiotic therapy consistent with the national antibiotic guidelines, or
2. A prescription of an antibiotic therapy considering the laboratory results including antibiotic susceptibility testing results.

Then, a metric will be used to describe the number and the proportion of the adapted antibiotic prescriptions.

Microbiological outcomes include a description of the bacteria isolated by culture, bacterial resistance rates as well as the delay between the sample being collected and the laboratory results communicated to the prescriber (in days and hours, online supplemental materials 1 and 2).

Clinical outcomes include mortality defined as death occurring during the patient's hospital stay or the length of stay defined as the duration between the admission date and discharge date from the hospital (online supplemental materials 1 and 3).

### Setting and laboratory capacity

All laboratories participating in the TSARA project are part of the nationwide RESAMAD network in Madagascar.[16] Description of the infrastructure in each hospital included in the TSARA project is presented in online supplemental material 4.

These laboratories perform diagnostics and AST for infectious diseases, and have the capacity to perform phenotypic confirmation of the presence of resistance. To participate in TSARA, laboratories have to meet minimal quality standards requirements defined by the RESAMAD network and participate in an external quality assessment programme. The laboratories also have the required infrastructure, equipment, supplies and resources to perform AST following EUCAST standards. The reporting of results to clinicians is made by qualified and trained technicians supervised by clinical microbiologists. The Mérieux Foundation performs regular training to laboratory staff and support the purchase of consumables (ie, Petri dishes, antibiotic disks) and reagents.

### Study population

The population of the study includes all patients (adults and children) arriving at the hospital who are inpatients and have bacterial samples taken by the physician and sent to the laboratory.

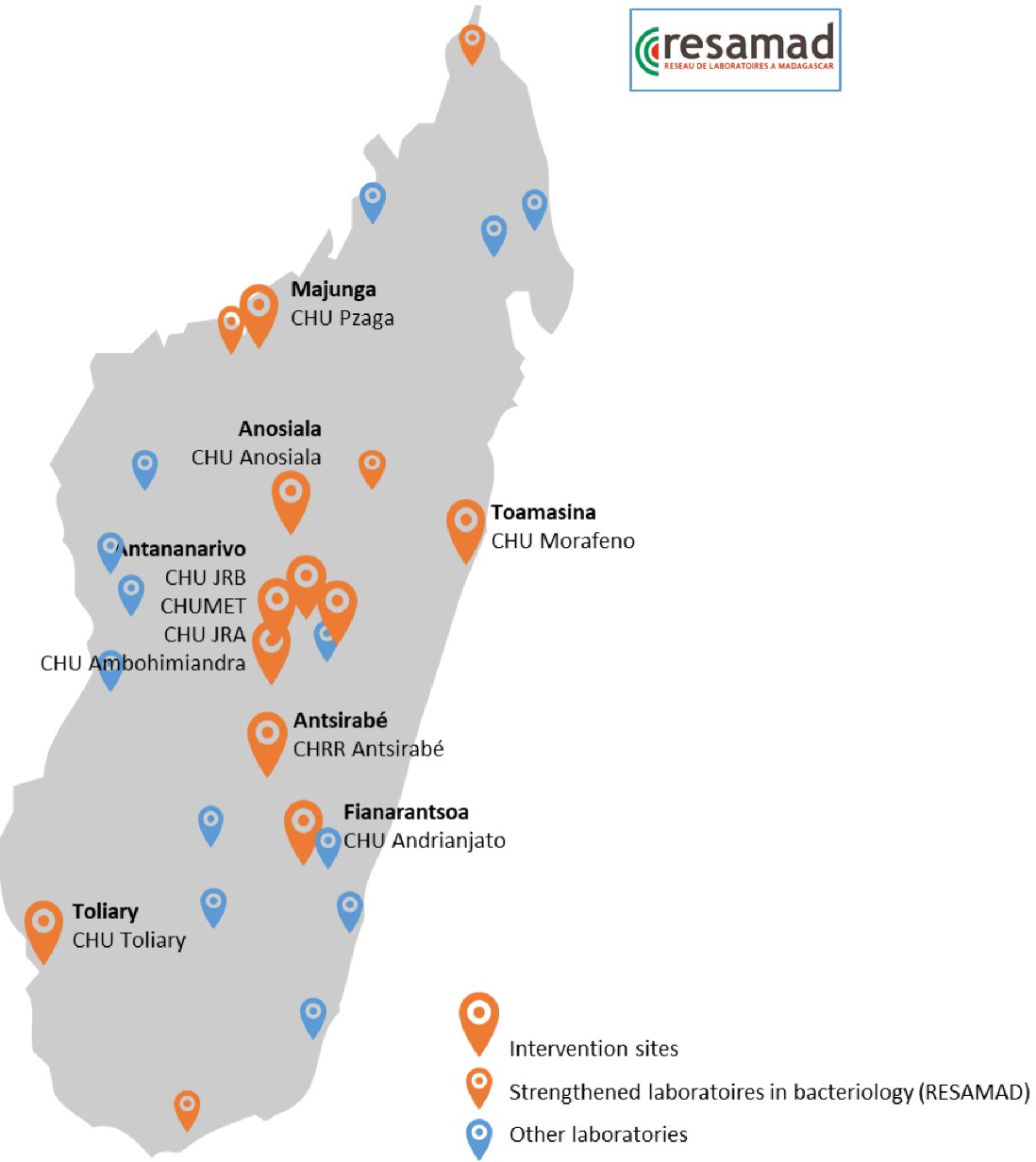

**Figure 1** Map of the laboratories and implementation sites of Technique de Surveillance Actualisée de la Résistance aux Antimicrobiens in Madagascar. RESAMAD, Réseau des Laboratoires à Madagascar.

### Inclusion criteria

All patients arriving at each hospital, who have a bacterial culture performed are included in the study. All positive and negative bacteriological samples from hospitalised patients in one of the participating institutions from January 2023 onwards will be included.

### Exclusion criteria

The following criteria will be excluded from the TSARA project:

1. All non-bacteriological specimens (virological, fungal, parasitic).
2. All samples from patients not hospitalised >24 hours (emergency, consultation, outpatient hospitalisation).
3. All environmental samples (water, air, surfaces, medical devices).
4. All postmortem samples.
5. All samples reported as contaminated are defined as results that do not contribute to clinical decision-making.
6. All specimens for which culture was not possible.
7. All specimens with laboratory non-compliance.
8. All the patients or legal guardians who object to be included in the TSARA project.
9. All patients with severe cognitive impairment.
10. All patients who do not have a full understanding of the French or Malagasy language.

### Participant timeline

Enrolled patients will be followed up during their hospital stay until discharge or death. Information on the vital status (alive, deceased, discharged or transferred to another hospital) will be collected.

## Patient and public involvement

Patients and the public were not involved in the design of this study.

## Data collection

Individual isolate-level data and AST on pathogens were collected routinely from the bacteriology laboratory using a first questionnaire (online supplemental material 1). These data were compiled using a second questionnaire with basic clinical, demographic and epidemiological information (including current antibiotic prescription) retrieved daily through face-to-face interviews with the patient and completed by medical records where necessary (online supplemental material 2). A third questionnaire was then filled in regarding follow-up data (online supplemental material 3). The data retrieved from TSARA meet the criteria and are linked to the GRAM project that seeks to estimate the burden of AMR worldwide.[15 17]

To enhance data collection and limit data entry bias, data were directly captured on a tablet connected to the Epicollect5 platform (https://five.epicollect.net). Data were collected by Malagasy investigators (medical interns, biologists or clinical research associates) using tablets dedicated to this project, each of whom received specific training on the TSARA methodology prior to the start of the study. The investigator will examine the bacteriological samples daily, or as frequently as possible. A new eligible sample corresponds to a new inclusion in the TSARA project.

## Statistical methods

### Sample size

TSARA is a surveillance project, as such there is no calculation on the number of subjects needed a priori. Statistical analyses will be adapted a posteriori according to the total number of inclusions during the project period and the power needed to obtain valid results.

## Description of the statistical methods

Any anomalies in the data will be checked with medical records during data cleaning. Data will be validated by quality control to identify outliers and inconsistencies by means of consistency tests (eg, filling in the inclusion criteria, date of admission to hospital later than the date of admission to the department, date of death later than the date of onset of symptoms).

Descriptive summary tables with numbers and frequencies will be drawn up. Similarly, visual representations of the variables may be made to detect irregular or missing values. Any changes to the data will be documented and recorded separately from the raw database. Recoded variables will also be filled in (eg, age categories).

A flow chart will be developed to describe the number of eligible, included and excluded patients and samples in our study. A descriptive analysis of the entire baseline population will be undertaken to identify the baseline characteristics of the study population (the specimen, bacterial, clinical and patient levels).

Categorical variables will be reported as frequencies and percentages of the total population. Quantitative variables will be described using the mean and SD if the distribution is normal, or the median and IQR if the distribution does not follow the normal distribution. The range with the minimum and maximum values will be filled in. For each variable, the proportion of missing values will be indicated.

Stratified analyses will also be provided to present results by facility, department, type of sample, and bacteria of interest (including species, and antibiotic resistance of interest) as well as by type of antibiotics prescribed. In addition, given the fact that data are collected for the adult and paediatric population, a stratified analysis will be performed for both populations.

A two-group comparison using univariate and then multivariate analysis and a logistic regression will be performed to estimate the associations between an adapted antibiotic therapy and the other variables. The OR and p value of each test will be reported. Only the variables with a p value lower than 0.2 will be retained to perform the multivariate analyses. In addition to the presentation of the multivariate analyses, the most efficient multivariate model will be presented after a top-down selection of only those variables with a p value less than 0.05. Stratified analyses will be conducted to identify potential confounders (comparison of crude and adjusted ORs) and interaction factors (Wald's test of homogeneity).

These analyses will be performed using the $\chi^2$ test or Fisher's exact test for qualitative variables, and the Student's t-test or the Mann-Whitney test (depending on the nature of the variable or the size of the sample) for quantitative variables. A significance level of less than 0.05 is considered associated.

Patient survival will be calculated from the date of enrolment until in-hospital death, based on the Kaplan-Meier method. Follow-up will be censored at 14 days and 28 days. Survival distributions will be compared by the log-rank test. Variables independently associated with survival will be identified with a Cox regression model based on relative hazard with a 95% CI.

## Missing data and outliers

Missing data will be documented individually. In the case of multiple missing data or if there is evidence of bias in the missing data on the variables of interest, multiple imputation methods may be used at the study level to substitute missing values. Imputation will be performed if the proportion of missing data does not exceed 20%. If this is the case, the variable will be removed from the statistical analysis.

A sensitivity analysis will be performed to compare the raw database (including missing data) with the imputed data.

Outliers were prevented by a blocking input mask, alert messages and an input guide. However, if new outliers are discovered after the quality checks and during the

analyses, they will be corrected as missing data. The scripts allowing these corrections will be communicated.

## Data management and archiving
### Case report form
The case report form only includes data necessary for participation in the TSARA project. All information required by the protocol is recorded in the anonymised case report forms. Data are collected as they are obtained in the laboratory and then at the patient's bedside, and recorded in these case report forms explicitly.

These case report forms are set up in each of the facilities using the electronic tablets equipped with Epicollect applications, including the TSARA project questionnaires. An instruction book to help the investigators use this tool was provided. The investigators are responsible for the accuracy, quality and relevance of all data entered, under the responsibility of the lead investigator.

These case report forms are associated with de-anonymised forms, in paper format, containing the first and last names of the patients enrolled, as well as an identification number shared between the paper and online forms. These second de-anonymised forms stay within the hospital and should allow possible corrections or deletions of data on request of the patients enrolled.

### Data management
Collected data are computerised by the epidemiologist in charge of the project. Electronic databases are anonymous and locked with a password known only by the scientific staff. These data will be kept for a minimum of 15 years after the end of the study.

### Archiving
The sponsor will keep the study documents (protocol and annexes, possible amendments, information forms, case report form, statistical analysis plan and output and the final study report) for a minimum of 15 years. After this period, the sponsor will be consulted before any data are destroyed. Study-related documents and reports may be subject to audit or inspection by the sponsor and/or other authorised bodies. No relocation or destruction will be made without the consent of the sponsor. At the end of the 15 years, the sponsor will be consulted for destruction. All data, documents and reports will be subject to audit or inspection.

### Confidentiality
All personal data on the study participants will remain strictly confidential. To respect their privacy and for confidentiality, all participant details will be anonymous for the purpose of database preparation. Study subjects are coded with four numerical numbers.

## Ethics and dissemination
### Ethics
The TSARA study was approved by the National Ethics Committee in Madagascar on 24 January 2023 (n°07-MSANP/SG/AGMED/CNPV/CERBM).

### Informed consent
Patients are fully and fairly informed, in understandable terms, of the objectives, their rights to refuse to participate in the study, or the opportunity to withdraw at any time.

All of this information is included on an information form given to the patient, and the patient's agreement will be collected by the investigator and will be recorded in the patient's source file.

The information will be communicated orally in French or in Malagasy by the investigators. An information form is also available in paper format in French and in Malagasy. When conditions permit, if the eligible patient does not speak French or Malagasy, the investigators will seek to volunteer a third party to provide a translation (a relative of the patient, or available hospital personnel).

### Regulatory compliance
The study will be conducted in accordance with applicable laws and regulations currently in place in Madagascar.

### Withdrawal criteria
Subjects may request to withdraw from the study at any time, without explanation, for any reason. In the event of premature exit, the investigator should document the reasons as completely as possible.

### Stopping the research study
The Mérieux Foundation reserves the right to discontinue the study at any time, if it is determined that the inclusion objectives are not met, or if the protocol is not respected in one or more participating institutions, or subject to other circumstances beyond the control of the Mérieux Foundation (eg, political instability). In case of premature termination of the study in one of the institutions, the information will be transmitted by the sponsor to the Malagasy interlocutors.

### Protocol amendments
In the eventuality of changes in the existing protocol that significantly affect the scope or the scientific quality of the investigation, an amendment containing a verbatim description of the changes and reference (date and number) to the submission that contained the original protocol will be submitted to the ethical committee for their approval.

### Dissemination
Results and scientific reports that emerge from this study will be made publically available. This is under the responsibility of the principal investigator in agreement with the associated investigators. Results will be reported following the guidelines from the Strengthening the Reporting of Observational Studies in Epidemiology (STROBE) consortium (www.strobe-statement.org) and STROBE-AMS.[18] Publication rules will follow international recommendations. The findings will also be shared with national health and sports authorities.

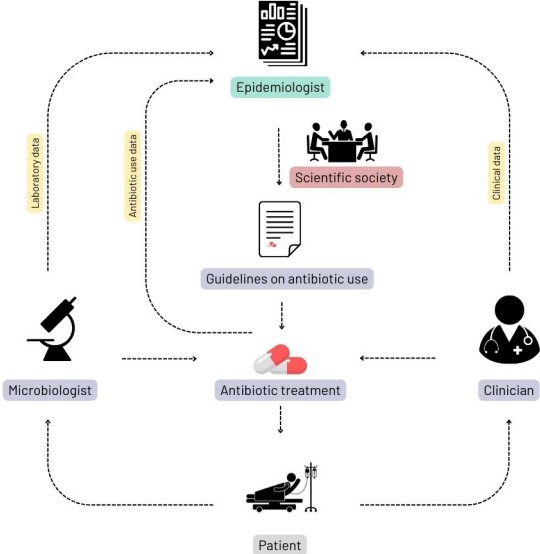

**Figure 2** Data flow between the different stakeholders.

Authorship will follow the guidelines established by the International Committee of Medical Journal Editors (https://www.icmje.org), which require substantive contributions to the design, conduct and interpretation and reporting of an epidemiological study.

These data will help the Malagasy scientific societies of infectious diseases (SPIM), paediatrics (SMP) and microbiology (SOMABIO) to tailor antibiotic recommendations according to the resistance patterns as presented in figure 2.

## DISCUSSION

The TSARA project will provide original results that could (i) allow a more precise epidemiological knowledge of bacterial infections, prescribed antibiotic therapies and local resistance levels; (ii) inspire, correct and tailor local practices, including recommendations and guidelines for empiric antibiotic therapy according to local resistance patterns; (iii) encourage communication and use of AST results with and between clinicians; (iv) allow monitoring of hospital antibiotic prescription (quantity, quality and spectrum, concordance with bacteriological results) and (v) provide evidence to improve laboratory services and interactions with clinicians.

Surveillance is the primary strategy for tracking emerging drug resistance in the population, and thus provides a tool to pick up any unusual organisms early and perform appropriate action. Regular dissemination of AMR data is key to elaborate local, evidence-based recommendations and revise the AMR national policy.

### Shortcomings

Implementing TSARA as an AMR laboratory and patient-based surveillance system presents some challenges. First, all the patients included in the study were hospitalised, and are self-paying for the majority of healthcare costs which induces a selection bias and may hamper the representativeness of the data. Mainly tertiary care hospitals belonging to the RESAMAD laboratory network were included due to their laboratory capacities, secondary and primary care centres were under-represented in TSARA. Consequently, results from the AMR data generated might be inflated as these facilities harbour more vulnerable and sick patients, who may have previously taken antibiotics and been in hospital and therefore be subject to higher antibiotic exposure. Additionally, clinicians may prescribe antibiotics in the absence of microbiological documentation or for the treatment of non-bacterial infections, even though such practices could contribute to elevated rates of antibiotic resistance. This tool, although innovative, is highly dependent on the capacity and the willingness of staff, on laboratory capacity, which can sometimes be heterogeneous within the hospitals and over the time, depending on the turnover of the staff and the availability of consumables in country. The TSARA project can be implemented only in a facility where laboratory services are functional, robust and where bacterial cultures and AST are performed routinely. In addition, patient treatment could be influenced by a delay in obtaining the culture results. In Madagascar, as well as many LMICs, it is common to use diagnostic microbiology testing after treatment failure to improve on broad-spectrum antibiotics,[19] which also overestimates local AMR rates. Negative perception of the laboratory capacity and associated costs could explain the unwillingness to perform microbiology testing by the clinicians.

No clear standard definition of multidrug-resistant bacteria has been set up by Malagasy health authorities, which results in difficulties in the data interpretation and comparison. Data collection currently relies on paper-based data capture, where no electronic medical records currently exist in the facilities. The quality of data may be another obstacle to getting validated results and there might be exposed to intersite variability. Clinical data and follow-up data could also be subject to a substantial proportion of missing data, as these are harder to retrieve than microbiological data. Finally, the investigator may have a lack of motivation to collect the data as it could be perceived as added work with a limited incentive or reward.

### Opportunities

The culture to collect of laboratory and clinical data is a serious challenge in LMICs. In 2021, the Mérieux Foundation implemented the Lab Book 3.0,[20] a Laboratory Information System which helps to computerise clinical laboratory data, in all the RESAMAD laboratories. This has both improved data collection and ensures data quality.

Additionally, the use of information and technology through a digital tool could be the next step for TSARA. It would encourage a timely reporting of AMR and clinical data ensuring the correct patient treatment. Thus, it may incentivise scientific societies to adjust national antibiotic guidelines according to the local resistance patterns and

hence support prescribers to make better, more informed clinical decisions and antibiotic prescriptions. Given the expansion of e-Health and connected devices, we imagine this tool to be integrated in a prescription assistance device on a mobile application as a future solution to combat AMR. Presenting the clinician with the antibiotic therapy recommendation associated with the WHO AWaRe classification has already shown its deployment in the fight against AMR.[6]

Furthermore, starting from January 2022, a change in the patient payment has occurred where any patient admitted to a Malagasy hospital with a life-threatening disease has received all the initial hospital care at no cost. Leadership and financial commitment from local health authorities has been invaluable (ie, direction of epidemiological surveillance and response of the Ministry of Health), and is key to the sustainability of the project and its expansion across the country.

Thanks to TSARA, significant progress should be made in AMR surveillance and antimicrobial stewardship in Madagascar. This project will also participate in supporting the implementation of the national action plan on AMR in Madagascar. Assessing the impact and the efficiency of this project that is, monitoring antibiotic prescriptions and checking their compliance with recommendations still need to be explored as clinicians' behaviours also drive AMR. Despite some challenges, future opportunities are encouraging for TSARA but rely on financial investment and strong leadership from health authorities to warrant the added value, the sustainability and the potential expansion of this project.

**Author affiliations**
$^1$Service Hygiène et Epidémiologie, Hospices Civils de Lyon, Lyon, France
$^2$Public Health, Epidemiology & Evolutionary Ecology of Infectious Diseases (PHE3ID) team, Centre International de Recherche en Infectiologie (CIRI), Inserm U1111, CNRS UMR5308, ENS de Lyon, Université Claude Bernard Lyon 1, Lyon, France
$^3$Direction des Opérations Internationales, Fondation Mérieux, Lyon, France
$^4$Service de Biologie, Hôpital Befelatanana, Antananarivo, Madagascar
$^5$Département de Santé Publique, Faculté de Médecine de l'Université d'Antananarivo, Antananarivo, Madagascar
$^6$Service des Maladies Infectieuses, Hôpital Befelatanana, Antananarivo, Madagascar
$^7$Centre for Neonatal and Paediatric Infection, St. George's, University of London, London, UK

**Acknowledgements** The authors thank all the biologists, namely Dr Christian Rafalimanana, Dr Lalaina Rahajamanana, Dr Zakasoa Ravaoarisaina, Dr Tiana Andry Razafinikasa, Dr Lalaina Raoelina, Dr Ainamalala Catherine Razafindrakoto, Dr Solotiana Rivo Rakotomalala, Dr Jocia Fenomanana, Dr Emile Ravelomandranto, Dr Irène Rakotoniaina and the local investigators of the RESAMAD network for their participation in the TSARA project. The authors are very thankful for Luciana Rakotoarisoa and Johannie Rakotoniaina for their support in the implementation of the project locally.

**Contributors** CE, MRaad, F-XB, LR, MRaberahona, SR, MRandria designed the study. CE and MRaad drafted the manuscript. CE, SR, AHR, VA and LR participated in the acquisition of data. CE, MRaad, SR, AHR, VA, MRaberahona, CEM, MRandria, LR, PV, F-XB have revised and approved the final version of the manuscript.

**Funding** The TSARA project has received funding from:Fondation Mérieux. PFIZER ISID Research Grant n°63019455.

**Map disclaimer** The inclusion of any map (including the depiction of any boundaries therein), or of any geographic or locational reference, does not imply the expression of any opinion whatsoever on the part of BMJ concerning the legal status of any country, territory, jurisdiction or area or of its authorities. Any such expression remains solely that of the relevant source and is not endorsed by BMJ. Maps are provided without any warranty of any kind, either express or implied.

**Competing interests** None declared.

**Patient and public involvement** Patients and/or the public were not involved in the design, or conduct, or reporting, or dissemination plans of this research.

**Patient consent for publication** Not applicable.

**Provenance and peer review** Not commissioned; externally peer reviewed.

**ORCID iDs**
Christelle Elias http://orcid.org/0000-0002-6962-161X
Mihaja Raberahona http://orcid.org/0000-0001-8857-5834

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
