## [Reviewer comments · BMJ Open]

ARTICLE DETAILS

TITLE (PROVISIONAL)	Implementation of an antibiotic resistance surveillance tool in Madagascar, the TSARA project: a prospective, observational, multicenter, hospital-based study protocol.
AUTHORS	Elias, Christelle; Raad, Mathieu; Rasoanandrasana, Saida; Raherinandrasana, Antso Hasina; Andriananja, Volatiana; Raberahona, Mihaja; Moore, Catrin; Randria, Mamy; Raskine, Laurent; Vanhems, Philippe; Babin, François-Xavier

VERSION 1 – REVIEW

REVIEWER	Barbieri, Elisa University of Padua
REVIEW RETURNED	09-Oct-2023

GENERAL COMMENTS	Dear Author, Dear Editor, Thank you for allowing me to review this protocol. It is not clear to me if this protocol is about an AMR tool development or an AMR tool implementation. The authors do not explain how this tool will be implemented in the case. This is vital to discriminate between an AMR surveillance protocol and an ASP implementation protocol. Moreover, it is not clear how this tool will help clinicians in prescribing antibiotics. Antibigram-driven therapy alone cannot be considered an ASP, but further actions are needed. Please find below some comments I believe would improve the manuscript. According to IDSA, “antimicrobial stewardship refers to coordinated interventions designed to improve and measure the appropriate use of antimicrobials by promoting the selection of the optimal antimicrobial drug regimen, dose, duration of therapy, and route of administration.” I think the title of this manuscript is misleading because in this paper Authors are describing an AMR surveillance tool implementation, not an ASP implementation. I suggest changing it to something as follows “Implementation/Development of an antimicrobial resistance (or susceptibility) surveillance tool in Madagascar, the TSARA project: a prospective, multicenter, hospital-based study protocol.” Please confirm that the design of the study. Is it an observational study? A quasi-experimental study? A before and after study? An interventional study based on the TSARA tool implementation? The protocol is too vague on the metrics that will be used to describe the changes in prescribing practices. Please specify.
--

	“Clinical outcomes include mortality or length of stay in hospital defined as....” Please complete. Microbiological outcomes include a description of the bacteria isolated by culture, bacterial ANTIMICROBIAL resistance rates results as well as the delay between the sample being collected and the laboratory results communicated to the prescriber. Please describe how you plan to evaluate the delay in time (hours/days/etc) You should evaluate outcomes separately for the adult and for the pediatric population. How will co-infections be evaluated? Will those samples be included in the tool? Please describe how you will define a contaminated sample. Figure 2 does not give added value to the description in the text. Which information are Authors willing to gather stratifying the analysis based on the antibiotics prescribed? The outcomes should be contextualized based on the antibiogram results. Please confirm all comparisons performed will be a two-group comparison. If not, please describe the appropriate statistical tests that will be used.
--	--

REVIEWER	Saison, Julien Centre Hospitalier de Valence
REVIEW RETURNED	29-Nov-2023

GENERAL COMMENTS	First, I would like to precise that this is a very interesting and well-written article, describing the implementation of a realistic and ambitious antibiotic stewardship program in Madagascar, where such programs are urgently needed. However, I have few questions:  - Table 1, page 5: could you precise if the data presented comes from the personal experience of the Malagasy authors? And/or by existing source or publication? - Page 6: Is there actually such a Malagasy national action plan on AMR? If any, to be implemented in the bibliography - Supplementary material 3: Please explain the choice of only 3 follow up data - Page 15: “data are collected” instead of "data is collected"
--

VERSION 1 – AUTHOR RESPONSE

Reviewer: 1
Dr. Elisa Barbieri, University of Padua
Comments to the Author:
Dear Author,
Dear Editor,

Thank you for allowing me to review this protocol. It is not clear to me if this protocol is about an AMR

tool development or an AMR tool implementation. The authors do not explain how this tool will be implemented in the case. This is vital to discriminate between an AMR surveillance protocol and an ASP implementation protocol. Moreover, it is not clear how this tool will help clinicians in prescribing antibiotics. Antibiogram-driven therapy alone cannot be considered an ASP, but further actions are needed. Please find below some comments I believe would improve the manuscript.

1. According to IDSA, “antimicrobial stewardship refers to coordinated interventions designed to improve and measure the appropriate use of antimicrobials by promoting the selection of the optimal antimicrobial drug regimen, dose, duration of therapy, and route of administration.” I think the title of this manuscript is misleading because in this paper Authors are describing an AMR surveillance tool implementation, not an ASP implementation. I suggest changing it to something as follows “Implementation/Development of an antimicrobial resistance (or susceptibility) surveillance tool in Madagascar, the TSARA project: a prospective, multicenter, hospital-based study protocol.”

Thank you for this comment in line with the Editor’s comment. TSARA is an AMR surveillance tool that gathers microbiological and clinical data (including antibiotic prescription data). To be consistent with the design and the objective of the TSARA project, the title has been modified accordingly as follows: Implementation of an antibiotic resistance surveillance tool in Madagascar, the TSARA project: a prospective, observational, multicenter, hospital-based study protocol.

The authors would like to emphasize that collecting data in a low-resource country poses a significant challenge, and providing any form of data represents a substantial stride towards further intervention against AMR. The authors argue that the IDSA definition of an ASP program, originating from an American scientific society, may not be entirely applicable to a resource-limited country like Madagascar.

2. Please confirm that the design of the study. Is it an observational study? A quasi-experimental study? A before and after study? An interventional study based on the TSARA tool implementation? TSARA is designed as an observational and prospective study, aiming to integrate microbiological data with clinical information. The TSARA project does not involve any planned interventions. Furthermore, the authors explicitly indicated in the statistical methods section that the TSARA project is a surveillance.

3. The protocol is too vague on the metrics that will be used to describe the changes in prescribing practices. Please specify.

Thank you for this comment. The authors have provided additional details on outcome metrics, explicitly defining the behavioural outcome as an adapted antibiotic prescription, characterized by either:

1. A prescription of an antibiotic therapy consistent with the national antibiotic guidelines, or
2. A prescription of an antibiotic therapy considering laboratory results including antibiotic susceptibility testing results.

4. “Clinical outcomes include mortality or length of stay in hospital defined as....” Please complete.

Thank you for this comment. Mortality is defined as death occurring during the patient's hospital stay. The length of stay is defined as the duration between the admission date and the discharge date from the hospital.

The methods section has been amended accordingly.

5. Microbiological outcomes include a description of the bacteria isolated by culture, bacterial ANTIMICROBIAL resistance rates results as well as the delay between the sample being collected and the laboratory results communicated to the prescriber. Please describe how you plan to evaluate the delay in time (hours/days/etc)

Thank you for this comment. As part of the microbiological questionnaire, the authors gather data on both the date of the sample and the date of delivery of the results to the prescriber. The delay will be

evaluated in days, as both dates are recorded in days, as depicted in Annex 1. The median or mean will be estimated based on the normal distribution of the delay. This estimation of the delay between the date of the sample and the date of delivery addresses one of the objectives related to laboratory quality assessment. Additionally, collecting information on the delay will also document the factors contributing to the non-adaptation of empiric antibiotic therapy or the delay for the clinician to receive the results of the bacterial culture and AST. In addition, determinants of a long delay could be assessed using survival methods.

6. You should evaluate outcomes separately for the adult and for the pediatric population.

Thank you for this comment. The authors included a sentence in the methods section regarding a stratified analysis according to the age of population (adult vs pediatric).

7. How will co-infections be evaluated? Will those samples be included in the tool?

Thank you for this comment. The TSARA project enabled to collect multiple bacterial isolates and samples for one patient. Bacterial co-infections can be documented as part of the TSARA project. Descriptive analysis of the bacterial co-infections can be performed as part of the study. However, TSARA did not aim to retrieve virological, fungal or parasitic samples so non-bacterial co-infections are not recorded.

8. Please describe how you will define a contaminated sample.

Thank you for this comment. A contaminated sample is characterized by a bacteriological outcome that does not align with or contribute to clinical decision-making. For example, microbiologists would consider a urine sample as contaminated due to the presence of a polymicrobial flora.

The methods section has been modified accordingly.

9. Figure 2 does not give added value to the description in the text.

Thank you for this comment. However if the reviewer agrees, the authors would like to keep this table as it gives visibility to the facilities participating to TSARA, which included relevant data on the size of the hospital, the location and the type of hospital.

10. Which information are Authors willing to gather stratifying the analysis based on the antibiotics prescribed? The outcomes should be contextualized based on the antibiogram results.

Thank you for this comment. Stratifying by the type of prescribed antibiotic (therapeutic class or ATC Code, for instance) would serve as a method to present the results and identify which specific molecule or therapeutic class is more likely to be adapted, aligning with the guidelines and/or considering laboratory results, including AST results as defined in the TSARA outcomes.

11. Please confirm all comparisons performed will be a two-group comparison. If not, please describe the appropriate statistical tests that will be used.

Thank you for this comment. The authors confirm that all comparisons will be a two-group comparison. The authors have added a sentence to specify this in the methods section.

Reviewer: 2

Dr. Julien Saison, Centre Hospitalier de Valence

Comments to the Author:

First, I would like to precise that this is a very interesting and well-written article, describing the implementation of a realistic and ambitious antibiotic stewardship program in Madagascar, where such programs are urgently needed.

However, I have few questions:

1. Table 1, page 5: could you precise if the data presented comes from the personal experience of the Malagasy authors? And/or by existing source or publication?

Thank you for this comment. The authors raise the fact that the references have been included in the Table 1 when available. Otherwise the information integrated in the SWOT analysis are based from the own experience of all the co-authors of the manuscript.

2. Page 6: Is there actually such a Malagasy national action plan on AMR? If any, to be implemented in the bibliography

Thank you for this comment. There is a national action plan on AMR existing since 2019. It has been included as a reference in the bibliography (reference n°16).

3. Supplementary material 3: Please explain the choice of only 3 follow up data

Thank you for this comment. The objective of the TSARA project is to evaluate behavioural changes in antibiotic prescribing practices. Consequently, the authors opted to limit data collection to the duration of the hospital stay. Nevertheless, recognizing the significance of outcome data, such as mortality and length of stay, particularly in the context of addressing the burden of AMR, the authors made the decision to incorporate limited information regarding patient follow-up. Furthermore, as TSARA aligns with the objectives of the GRAM project of the University of Oxford, which aims to estimate the global burden of AMR, the inclusion of restricted follow-up data was considered essential. The authors justified this choice, emphasizing that expanding follow-up data collection would have hampered the overall efficiency of TSARA's data collection process and posed a greater time burden on surveyors.

4. Page 15: "data are collected" instead of "data is collected"

Thank you for raising this typological error. The sentence has been modified accordingly.

VERSION 2 – REVIEW

REVIEWER	Barbieri, Elisa University of Padua
REVIEW RETURNED	07-Feb-2024

GENERAL COMMENTS	Dear Authors, I believed that the quality of the manuscript improved after addressing the points raised. There are a couple of points remained to address. Please find them below Mortality is usually assessed at a defined time-point, i.e. 14 days, 30 days after the day of the hospitalization start. Please define this in the protocol. I would advise the authors to collect data on the time to assess the delay between the sample being collected and the laboratory results communicated to the prescriber in hours, not in days. Indeed days might be useful in the short time to assess huge differences in delay, but hours are more precise and more useful in the long time when the surveillance tool is implemented.
--